# Analysis of the Impact of Selected Physical Environmental Factors on the Health of Employees: Creating a Classification Model Using a Decision Tree

**DOI:** 10.3390/ijerph16245080

**Published:** 2019-12-12

**Authors:** Miriam Andrejiová, Miriama Piňosová, Ružena Králiková, Bystrík Dolník, Pavol Liptai, Erika Dolníková

**Affiliations:** 1Department of Applied Mathematics and Informatics, Technical University of Košice, 040 01 Košice, Slovakia; miriam.andrejiova@tuke.sk; 2Department of Process and Environmental Engineering, Technical University of Košice, 040 01 Košice, Slovakia; miriama.pinosova@tuke.sk (M.P.); ruzena.kralikova@tuke.sk (R.K.); 3Department of Electric Power Engineering, Technical University of Košice, 040 01 Košice, Slovakia; bystrik.dolnik@tuke.sk; 4Institute of Recycling Technologies, Technical University of Košice, 040 01 Košice, Slovakia; 5Department of Building Construction, Technical University of Košice, 040 01 Košice, Slovakia; erika.dolnikova@tuke.sk

**Keywords:** physical environmental factors, occupational health, decision tree model, questionnaire survey, microclimate

## Abstract

During the process of designing and implementing a working environment, there is a need to guarantee adequate conditions for future workers’ health and well-being. This article addresses the classification of employees characterized by several basic input variables (gender, age, class of work). The investigated variable was the health of employees. This article aims to create a prediction classification model using the classification tree, which can be used to classify new cases into appropriate classes as accurately as possible. Objective measurements of microclimatic parameters were performed by the Testo 435 instrument. The subjective evaluation was performed by a questionnaire survey formed from the training group of 80 respondents and independently verified by the test group of 80 more respondents. The result confusion matrix shows that the number of correctly classified respondents was 69 from a total of 80 respondents. The overall accuracy was $A_{C}=0.863$, which means that the likelihood that respondents are properly classified in the correct health class is 86.3%. Based on the model obtained using the classification tree, we can classify respondents into the relevant class for their state of health. The respondent is classified into the class of work for which particular health and working conditions are most likely.

## 1. Introduction

The human perception of the environment depends on four basic factors namely the thermal environment, lighting, acoustics, and indoor air quality. These factors are related both to Occupational Health and Safety and to Indoor Environmental Quality, representing complementary aspects, but they are characterized by two different approaches. Human exposure to the hazardous conditions is governed by national and international directives where there limit values are indicated for each factor. The exposure to hazardous agents is addressed based on pass/fail criteria. However, while compliance with these limits ensures safe working conditions, it does not imply the achievement of conditions of comfort. The perception of comfort has long been studied by the scientific community, which has analyzed the relationship between the perception of comfort and environmental conditions. There is enough knowledge of the thermal environment available. The International Standards Organization has processed sufficient quantities of legislation in this area. However, given the over-abundance of information available, it is not always easy for practitioners to choose the method of evaluation best-adapted to conditions in the workplace that is to be studied.

Quality of the working environment is one of the most important factors affecting the performance of employees, their well-being, reduced workload, reducing errors, incapacity for work, or the emergence of occupational diseases. Achieving the right balance in workplaces is not always easy. The industrial work environment is generally affected by one or more undesirable influences [1]. Lumnitzer et al., in his publication [2], states that the thermal balance of the body and at the same time thermal comfort are influenced by the factors: indoor environment (air temperature, relative humidity of air, radiant effects of surrounding areas, airflow velocity and turbulence), the body exposed (metabolic value, clothing), additional factors (acclimatization to outdoor climate, acclimatization to indoor climate, body and subcutaneous fat, age and gender).

Evaluation of the working environment from the viewpoint of microclimatic conditions is an important topic, which is of great interest from the research point of view. McIntyre and Griffiths [3] researched, by using the questionnaire method, the response of people aged 16–19 years who were exposed over a period of 6 h to three levels of relative humidity: 20%, 50% and 75% at temperatures of 23 ${}^{\circ }C$ and 28 ${}^{\circ }C$. The temperature parameters were recorded at hourly intervals. Andersen et al. [4] studied the subjective perception of humidity and temperature in 48 young male subjects exposed to clean air at 23 ${}^{\circ }C$ with relative humidity of 10%, 30%, 50%, and 70%. The results of their study indicated that the decrease and rise of relative humidity caused highly significant changes in the objective sensation of temperature, even though the temperature parameter was held constant throughout. Wyon et al. [5] monitored a group of 36 males and the same number of females 17-year-old subjects in standard cotton dress (clothing insulation 0.7 clo) who were exposed in groups of 4 to patterns of changing the air temperature. Temperatures remained within the range of 20 ${}^{\circ }C$ to 29 ${}^{\circ }C$ and did not increase more rapidly than 4 ${}^{\circ }C$
$h^{-1}$. Each individual recorded his or her thermal sensation using a dial voting apparatus. Authors found that there were significant differences between the responses of males and females, with males in general feeling hotter and reacting more rapidly to changes in temperature. Erlandson et al. [6] in two Australian cities, performed a questionnaire study of 1771 administrative workers considering thermal comfort in air-conditioned offices. Mann-Whitney analyses (after correction for climatic variables) showed that subjects with higher job satisfaction had thermal sensations closer to “neutral”. The combination of objective and subjective evaluation of thermal comfort conditions has been studied by Eide et al., Kuchen and Fisch, Della Crociata et al., Kosala et al., and Morgado et al. [7,8,9,10,11]. From the perspective of objective evaluation of thermal comfort of monitored subjects is, in the article from Lu et al. [12], described interesting way to apply infrared thermography to develop individual human thermal models to predict thermal sensations.

Several contributions have already addressed the use of various mathematical and statistical methods in the processing of data from various research areas. According to Witten et al. [13], mastering knowledge from databases can be defined as the non-trivial acquisition of implicit, unknown, and potentially useful information from data. We can understand data-mining more simply as a method to acquire hidden but useful information from a vast quantity of data. The methods used in data-mining include, for example, techniques of classical statistics, (regressive analysis, logical regression), decision trees, association rules, clustering, neural networks, etc. Kalyankar et al. [14] used machine learning and data mining to analyze data on patients with diabetes. Khateeb and Usman [15] predicted heart disease using a K-nearest neighbor classifier, which is a data-mining technique. Neural networks were used to predict heart disease using risk factors in the paper [16].

This article aims to create a prediction classification model, which can be used to classify new cases into appropriate classes as accurately as possible.

## 2. Materials and Methods

The issue of the heat-humidity microclimate is addressed in Slovakia by several decrees, government regulations, and several STN standards (taken from international standards). The objectification of the physical quantities of the heat-humidity microclimate is carried out following the Technical Guideline of the Ministry of Health of the Slovak Republic no. 16/2013 [17], which regulates the procedure for the measurement and assessment of the heat-humidity microclimate. In order to assess the fulfillment of heat-humidity microclimate requirements at work, the basis is the results of direct or indirect measurement, and their comparison with the values of parameters set out in Decree no. 99/2016 of the Ministry of Health, which is a regulation implementing Law no. 355/2007 on the protection, promotion, and development of public health [18,19]. The minimum safety and health requirements for the workplace are laid down in Government Regulation of the Slovak Republic no. 391/2006 on minimum safety and health requirements for the workplace [20].

During objectification of the physical factors of the heat-humidity microclimate by measurement, current methodologies, and measuring techniques are used [21]. The evaluation is performed on the basis of the comparison of the measured values with the optimal, allowed limits. If the measured values of a factor exceed limit values, it is important to ensure to reduce the effects of this factor by applying different measures [22]. The strategy of risk assessment for prevention of stress or discomfort in mild, cold and hot working conditions is addressed in ISO 15265:2004 [23]. The companies, in which the measurements of microclimate parameters were performed and subsequently the questionnaire survey, were industrial plants engaged in the production of components for cars. All employees, that were subject to research, worked indoors. In the monitored workplaces were measured air temperature ($t_{a}$), relative humidity ($r_{h}$), and airflow velocity ($v_{a}$). These parameters were measured by the multifunctional Testo 435-2 instrument with the hot wire probe (ϕ12mm) for measurements of air velocity, temperature, and relative humidity. The temperature measuring range −20 ${}^{\circ }C$ to 70 ${}^{\circ }C$ (accuracy $\pm 0.3\;{}^{\circ }C$), the relative humidity measuring range 0–100% (accuracy ±2%) and the air velocity measuring range 0 $m\;s^{-1}$ to 20 $m\;s^{-1}$ (accuracy $\pm 0.03\;m\;s^{-1}$). The manufacturer is the company TESTO GmbH. Measurements were performed in several measuring places with a two-hour periodicity for 24 h and with the sampling rate of 15 s. Based on the measurement method used, the measuring instrument, the measuring conditions and the experience of the measurers, was established the measurement uncertainty of ($t_{a}\pm 0.2\;{}^{\circ }C$, $v_{a}\pm \left(0.05+0.05v_{a}\right)$, $r_{h}\pm 3.0\%$).

Subjective assessment methods are most often performed on the basis of a questionnaire survey to determine the subjective perception of the factor acting on the tested subject/person. The questionnaire survey was conducted in the spring of the year 2019 (in particular the months of March, April, and May) and we were interested in two main areas:Heat and humidity microclimate conditions in the workplace (satisfaction with humidity, temperature, and airflow at the workplace) need for increased humidity and air temperature.Workplace health problems of employees such as a feeling of fatigue during working hours, symptoms of spinal pain, headache, feeling cold symptoms, dry nasal mucous membranes.

The supplementary questions included information specifying data on gender, age, and the medical condition of the respondents, class of work, drinking regimen in the workplace, work clothing of staff, and time to rest.

Respondents were assigned to three classes of work 1a, 1c, and 2a. The classes of work by total energy expenditure, in the sense of [18], is defined as follows:**1a**—work in a position with minimal movement activity, or connected with light manual work with the arms and hands (Work1).**1c**—standing work occasionally connected with slow walking with carrying of light loads up to 10 kg or overcoming light resistance (Work2).**2a**—standing work with constant use of both hands, shoulders, and legs, or work associated with carrying loads up to 15 kg (Work3).

The respondents subjectively evaluated the thermal and humidity microclimatic conditions and health problems on an evaluation scale as follows: “never = 1”, “occasionally/rarely = 2”, “often = 3”, “very often/almost always = 4”.

## 3. Theory and Calculations

The evaluation of the experimental tests used basic methods of mathematical statistics and probability theory. The assessment of the independence between the two categorical variables was used Pearson’s Chi-square test of independence. The decision tree is used to create the classification model.

### 3.1. Pearson’s Chi-Square Test of Independence

Pearson’s Chi-square test of independence determines whether there is a statistically significant relationship between two categorical variables A and B. We tested the null hypothesis H${}_{0}$: *the monitored variables A and B are independent*, against the alternative hypothesis H${}_{1}$: *the monitored variables A and B are not independent.* The decision to reject or not reject the null hypothesis is made using the *p*-value. If the *p*-value <α then the null hypothesis is rejected in favor of the alternative. If the *p*-value ≥α, the null hypothesis is not rejected. To determine the degree of dependence of the categorical variables we chose Pearson’s contingency coefficient C${}_{p}$,
(1)$$C_{p}=\sqrt{\frac{\chi^{2}}{\chi^{2}+n}}\;,$$
where $\chi^{2}$ is the Pearson Chi-square statistic. The higher the coefficient, the closer the dependence between the variables.

### 3.2. Creation of the Classification Model Using a Decision Tree

Decision trees are an important tool for classification and prediction and to facilitate decision-making in various decision-making problems. Decision trees and decision rules are the basis for the application of data-mining methodologies to many real-world applications for a powerful solution to classification problems. Depending on the type of output (target) variable, we divide them into classification and regression decision trees. Based on the nature of the output variable, in this research, we use a classification tree. The purpose of classification is to create a classification model, which would enable appropriate values of the output variable to be assigned to the values of the input variables of an object. The purpose of a decision tree is to classify objects into classes (categories) of output variables.

There are several algorithms for creating a decision tree. An algorithm for creating a decision tree always starts with the training group, which consists of *n* objects. Each object is characterized by *k* input variables, i.e., A${}_{j}$, *j* = 1, 2, …, k. Input variables can be discrete or continuous. For each object the value of the target variable *Y*, which can take *m* different levels or classes. The key task is to select the appropriate variable for tree splitting. Various criteria can be used to find the most suitable variable for the splitting, such as Entropy, Information Gain and Information Gain Ratio, Gini Index, and Chi-Square test. In this paper, we use the C5.0 algorithm that is suitable for all types of attributes; however, the output variable is categorical. A splitting criterion is the Information Gain and Entropy. To create and visualize the decision tree was used the software R (package C5.0).

### 3.3. Verification of the Classification Model

The training group is used to create the classification tree and classification rules. The test group is used to verify the classification model. For the measurement of the correlation of the output variable, we will use the overall accuracy $A_{c}$,
(2)$$A_{c}=\frac{C-E}{C}=1-\frac{E}{C}=1-R\;,$$
where *R* is the total error rate, *E* is the number of incorrectly classified cases and *C* is the total number of cases in the group.

To identify the rate of agreement between the two classifications (the real classification of the cases and the classification of cases using the model) we will use Cohen’s kappa,
(3)$$\kappa =\frac{p_{o}-p_{e}}{1-p_{e}}\;,$$
where $p_{o}$ is the probability of successful classification (overall accuracy $A_{c}$) and $p_{e}$ is the probability of success due to chance.

The recommended scale is: less than 20% poor agreement, from 20% to 40% fair agreement, from 40% to 60% moderate agreement, from 60% to 80% good agreement, and more than 80% is very good/excellent agreement.

## 4. Results and Findings

Experimental research and processing of results were carried out to:analyze the microclimatic conditions in the workplace and the satisfaction of respondents with those conditions in the workplace,analyze the health status of respondents and determine the dependence of the health status of respondents on the selected factors,create a classification model for the health status of respondents using a classification tree based on the training group of respondents and verify its classification and predictive ability with the test group of respondents.

A total of 160 respondents aged from 23 to 60 years of age participated in the experimental research. In the group were 71 women (female) (44.4%) and 89 men (male) (55.6%). In the age group up to 30 years, there were 42 respondents (26.3%), and from 31 years to 40 years, there were 44 respondents (27.5%). In the 41–50 years age group, there were 41 respondents (25.3%), and in the group over 50 years, there were 33 respondents (20.6%). There were 30 respondents (18.7%) whose work was in class of work 1a (the lowest), 94 respondents (58.8%) with class of work 1c, and 36 respondents (22.5%) with class of work 2a. 56 respondents (35.0%) stated that they had no health problems, 79 respondents (49.4%) described their state of health as being with mild health problems. 25 respondents (15.6%) reported that their health was poor and they have serious health problems. To find a classification model, we divided the group into two parts: the training group and the test group. In both groups, there were 80 respondents.

Results of measured parameters particular air temperature ($t_{a}$), relative humidity ($r_{h}$), and airflow velocity ($v_{a}$) at selected 3 measuring points of the monitored workplaces for individual class of work are graphically summarized and depicted in Figure 1, Figure 2 and Figure 3.

### 4.1. Analysis of Thermal and Moisture Conditions in the Workplace

Only 11 respondents surveyed (6.9%) said that the humidity in the room where they work was always suitable for them. 67 respondents (41.9%) often, and 75 respondents (46.9%) rarely, found the humidity in the room where they work suitable for them. An increase in the humidity of the air was wished for sometimes by 99 respondents (61.9%), and often by 38 respondents (23.8%). As many as 82 respondents (51.3%) are rarely satisfied with the circulation rate of air in the room. The circulation rate is satisfactory for 61 (38.1%) respondents. The air temperature in the room was suitable often or very often for 74 respondents (46.3%). And 77 respondents (48.1%) said they are rarely satisfied with the set temperature of the room where they work. 101 respondents (63.1%) do not wish for there to be a higher temperature in the room. 45 respondents (28.1%) rated their comfort with the thermal conditions as feeling warm, and 54 (33.8%) as moderately warm. 49 respondents (30.6%) identified their feelings as neutral, 8 (5.0%) as moderately cold and only 4 respondents (2.5%) reported feeling cold.

Respondents were also asked about the kind of work clothes which they use in performing their work. Ordinary working clothes were used by 146 respondents (91.3%) and only 14 respondents (8.7%) needed special prescribed work clothes for their job (e.g., waterproofs, various types of aprons, etc.). A drinks regimen was almost always ensured for only 94 respondents (58.8%), often for 57 respondents (35.6%) and only rarely for 9 respondents (5.6%). In the event of prolonged work (more than 4 h), only 83 respondents (51.9%) almost always have the option to leave the room where they work during breaks. 60 respondents (37.5%) have such a possibility often and 17 respondents (10.6%) very rarely.

In the next part of the research, we were interested in the relationship between the selected variables (α=0.05). When evaluating, we used the following labels for variables: gender of the respondent (A1, Gender), age (A2, Age), class of work (A3, Work), satisfaction with the humidity in the room (B1), increased humidity in the room (B2), satisfaction with the airflow in the room (B3), satisfaction with the air temperature (B4), increase in the air temperature (B5), evaluation of feeling related to thermal conditions (B6). The graphical representation of the subjective evaluation of heat and humidity conditions in the working environment according to the surveyed groups (training, test) is shown in Figure 4.

First, we tested the dependence of two variables: gender of the respondent (A1) and satisfaction with microclimate conditions (B1–B6). The results of the Pearson Chi-squared test of independence show that dependence was found between the gender variable and the variable of the respondents’ satisfaction with humidity of the air in the room (B1, $\chi^{2}=8.36$, *p*-value = 0.039<α). For the assessment of the degree of dependence, we used Pearson’s coefficient of contingency $C_{p}$. From the calculated values it follows that there is a weak dependence ($C_{p}=0.223$). On the other hand, dependence was not confirmed between the variable gender of respondents (A1) and satisfaction with the temperature of the air in the room (B5, *p*-value = 0.656). During testing it was found there was a moderate correlation between the gender of the respondents and their satisfaction with the circulation of air in the room (B3, *p*-value = 0.010), and between the gender of the respondents, and increasing humidity (B2, *p*-value = 0.002). Analogously, we investigated the dependence of the other two variables A2 and A3 on variables B1 to B6. The results, for 160 respondents (n=160), are presented in Table 1.

### 4.2. Analysis of the Health Condition of Respondents

During working hours, only 33 (20.6%) of respondents do not feel tired. On the other hand, 93 (58.1%) sometimes feel tired and 32 respondents (20.0%) often feel tired. For 47 (29.4%) respondents, the action of the airflow never causes local pain in the spine. Pain in the spine is suffered sometimes by 83 (51.9%) respondents and often by 27 (16.9%) respondents. 62 (38.8%) respondents do not suffer headaches during working hours. 81 (50.63%) respondents sometimes and 16 respondents (10.0%) often suffer from headaches. A feeling of getting a cold during working hours was not suffered by 48 respondents (30.0%) suffered sometimes by 93 (58.1%) respondents and often by 19 respondents (11.9%). The feeling of dried nasal mucosa during working hours is suffered sometimes by 91 (56.9%) and often by 31 (19.4%) respondents. The feeling of dried mucosa was never experienced by 37 (23.1%) respondents. Graphical representation of subjective assessment of health symptoms, depending on the studied group (training, test) is shown in Figure 5.

In this case we were interested in the relationship between selected variables: gender of the respondent (A1), age (A2), class of work (A3), feeling tired (B7), having pain in the spine (B8), headache (B9), feeling colds (B10) feeling of dried nasal mucosa (B11) and the incidence of health problems (B12). From the results of the Pearson chi-squared test of independence, dependence was found between the variable gender of respondents (A1) and the variable headache (B9, *p*-value = 0.048<α). From the calculated values it is indicated that this is a very weak dependence ($C_{p}=0.191$). The relationship was also confirmed between the variable age group (A2) and all health problems (B7 to B12). The resulting values of the Pearson coefficient $C_{p}$ show that it is a moderate dependence. The results, for 160 respondents (n=160), are presented in Table 2.

### 4.3. Classification Model of Health Status of Respondents

We base the determination of the classification model on the training group of respondents. To verify the classification and predictive ability of the model we use the test group of respondents.

The **training group** consists of 80 respondents. In the group, there are 48 men (60.0%) and 32 women (40.0%). There are 20 respondents (18.3%) in the up to 30 age group and there are 25 respondents (25.0%) in the category of 31–40 years of age, 18 (31.7%) in the 41–50 age group and 17 respondents (25.0%) over 50. There are 15 respondents (51.7%) with work in class 1a (Work1), 44 respondents (23.3%) with work in class 1c (Work2) and 21 respondents (25.0%) with work in class 2a (Work3). 29 respondents (36.3%) stated that they had no health problems. Up to 38 respondents (47.5%) described their health condition as having mild health problems. 13 respondents (16.3%) reported that their health is poor and they have serious health problems.

The **test group** consists of 80 respondents. In the group, there are 41 men (51.3%) and 39 women (48.7%). 22 respondents (27.5%) are in the age group of up to 30 years, 19 respondents (23.8%) are in the category from 31–40, 23 (28.7%) are in the age group of 41–50 years and 16 respondents (20.0%) are over 50. 15 respondents (18.8%) have work in class 1a (Work1), 50 respondents (62.4%) have work in class 1c (Work2) and 15 respondents (18.8%) have work in class 2a (Work3). 27 respondents (33.8%) stated that they have no health problems. And 41 respondents (51.3%) described their state of health as with mild health problems. 12 respondents (15.0%) reported that their health is poor and they have serious health problems.

Determining the classification model was based on the training of respondents. The classification criterion is the variable Health, which has three classes: Health1, Health2, and Health3. The other three input variables (Gender, Age, Work) characterize the individual respondents (Table 3).

The result of the decision tree is shown in Figure 6. All three input variables are used for any level of the tree (Age 100%, Work 75%, Gender 56.25%). The first division occurs based on the age category of the respondents. The subdivision is based on the class of work.

From the established decision tree, it follows that good health (Health1) is found in respondents who are in Age1 (regardless of gender and classification of work) and men with work in class Work1 or Work2 who are in Age2. Serious health problems (Health3) were found in respondents who have work in class Work3 and are in Age3 and Age4. A better state of health (Health2) was found in other cases.

To describe the agreement between the classification obtained from research and the classification determined by the decision tree we use a confusion matrix (Table 4). The comparison showed that the classification decision tree model has wrongly classified 11 respondents.

The confusion matrix shows that the number of correctly classified samples was 69 respondents from a total of 80 respondents. The overall accuracy was $A_{C}=0.863$, which means that the likelihood that respondents are properly classified in the correct health class is 86.3%.

For a correct assessment of the predictive force of the model, we moved to verification using the test group. The results of the confusion matrix for the case of the test group are in Table 5. In this case too, the classification model did not classify 100% of all respondents correctly in the target categories. The overall accuracy was $A_{C}=0.850$. The results of the classification model applied to the test group predicted the correct type classification of a respondent (85.0%) within particular health in the time.

The level of inclusion of a respondent into the same class of health status using the model and the real facts is also expressed using Cohen’s kappa coefficient. The formula shows that in the case of the training group, Cohen’s kappa value is equal to 0.77. In the test group, Cohen’s kappa value is equal to 0.74. In both cases, there is a good agreement of the classification of respondents. The results of the overall accuracy of the classification, Cohen’s kappa and its 95.0% confidence intervals are in Table 6.

Based on the model obtained using the classification tree, we can classify respondents into the relevant class for their state of health. The resulting classification of all possible cases, which are characterized by the variables (gender of the respondent, age, class of work) is in Table 7.

## 5. Discussion

### 5.1. Use of Decision Trees

The use of decision trees has wide use, as the following works suggest too. Dangare and Apte [24] use various data-mining techniques such as decision trees, Naïve Bayes, as well as neural networks to analyze heart disease. Vévoda et al. [25] monitored factors in the working environment influencing the decisions of nurses to stay with or leave their employer. The collected data from the questionnaires were analyzed using data-mining tools, decision trees, and non-parametric tests. Iheme et al. [26] used a naïve Bayesian classifier and a decision-stump-tree classifier to create a decision-making system to support child diagnostics in rural areas in India. Sumbaly et al. [27] used decision trees for cancer diagnosis and Kumar et al. [28] to analyze data from road traffic accidents.

### 5.2. Effects of Inappropriate Microclimate Conditions

In a warm environment or with rising production of metabolic heat, the human body responds by the widening of the subcutaneous vessels (vasodilatation), thus increasing the supply of subcutaneous blood. This increases skin temperature that will increase removal of heat from the body. If the increase in skin temperature cannot restore thermal balance, sweat glands are activated and evaporation cooling begins. However, if these two mechanisms cannot restore the body’s thermal balance, the body overheats—hyperthermia. The first health signs of hyperthermia are weakness, headache, nausea, short breathing, accelerated heart rate (up to 150 $\min^{-1}$), etc. In heat shock, the temperature rises rapidly over 41 ${}^{\circ }C$, sweating stops, coma starts and death occurs. In a cold environment, the human body first responds by lowering the subcutaneous blood circulation, reducing the temperature of the skin, which consequently reduces heat loss. This process is accompanied by the emergence of goose pimple or the atavistic phenomenon of hairs standing up on the skin, which results in better thermal insulation of the skin. If this mechanism is ineffective, muscle tension begins, shivering which increases the body’s thermal output. If these physiological responses do not provide thermal balance, the cooling of the body will occur—hypothermia and the internal temperature may fall below 35 ${}^{\circ }C$. If the temperature of the body begins to fall, the heart rate decreases and there is a disruption of blood circulation, with death occurring at a core temperature between 25 ${}^{\circ }C$ and 30 ${}^{\circ }C$. For these reasons, these health symptoms should not be neglected and great importance should be attached to them.

### 5.3. Prevention, Measures, and Limitations

Appropriate workwear for staff is an essential part of working in unsuitable heat-humidity conditions. When working in the cold, the thermal insulation of an employee is of utmost importance, i.e., protective clothing. Suitable clothing prevents the wind and moisture from penetrating the skin while enabling sweating and capturing sweat. Acclimatization to the cold is of great importance, i.e., the gradual prolonging of the stay in the cold working environment. Prevention includes instruction in using protective clothing and personal protective equipment, as well as recognizing early symptoms of general and local hypothermia and providing first aid.

When working in excessive heat, technical measures such as reducing the intensity of the heat source, removing the source (mechanical apertures, barriers), local employee cooling (air or water showers), general ventilation (natural, forced) and organizational measures among which we can include the regime of work and rest (rotation of employees, working breaks in favourable microclimatic conditions, etc.). We must also remember the drinking regimen of employees. Its goal is to replace the loss of fluid and salt during the working shift to avoid unwanted negative health symptoms.

The evaluation of the data obtained was based on the assumption that employees in their work positions respect safety and sanitary conditions at the workplace. Employees are trained in the use of personal protective equipment in the performance of their work. Also, are instructed in the possible negative effects on their health, when will not use them.

## 6. Conclusions

The effect of individual heat-humidity microclimatic factors on the working environment on human beings and their health can not be evaluated in isolation, because their mutual combinations can significantly influence, increase or multiply, depending on the work done.

The main objective of this study was to implement, evaluate and analyze a questionnaire survey of the subjective evaluation of the heat-humidity microclimatic conditions of the working environment in selected working positions. It is important to be aware that the lack of appreciation of some of the factors of the heat-humidity microclimate of the working environment results in a deterioration in working conditions, impairment of employee health, increased fatigue and overall discomfort to the employee, which may increase the risk of injuries. The positive effect of heat-humidity microclimatic conditions enables: the increased working capacity of the employees (good health), increased work performance of the employees by reducing the mental and physical load, increase the safety of work as well as an improvement the health of employees.

We are the first to have focused on the analysis of thermal and humidity conditions in the workplace or the health condition of respondents depending on the examined group. We observed the dependence between selected variables (A1–A3 and B1–B6), or (A1–A3 and B1–B12). In determining the classification model, we used a training group of respondents. We used a test group of respondents to verify the classification and prediction capabilities of the model. The Health attribute with its three classes (Health1, Health2, Health3) was chosen as the classification criterion. Other input variables of the classification model chosen were gender, age, class of work, which was used to characterize the individual respondents, while the target attribute was the health condition of respondents.

The results of the questionnaire survey may be a challenge for employers and management of organizations to implement preventive and corrective measures to adhere to the thermal and humidity microclimatic conditions in workplaces to meet the requirements set by legislation, regulations, recommendations or standards.

The model is based on experimental research: It is created based on the training group (80 respondents) and independently verified by the test group (80 respondents). From the results of the classification model implies several conclusions. We can assume that,
good health condition (Health1) is likely for employees up to 30 years of age, regardless of their gender and class of work,having more serious problems (Health3) is likely for employees over 50 years of age in class of work 2a regardless of gender, and female in the 41–50 age group in the same class of work,in all other cases, a moderate state of health is likely (Health2).

The confusion matrix shows that the number of correctly classified samples was 69 respondents from a total of 80 respondents. The overall accuracy was $A_{C}=0.863$, which means that the likelihood that respondents are properly classified in the correct class of work is 86.3%. Based on the model obtained using the classification tree, we can classify respondents into the relevant class of work, for their condition of health. The respondent is classified into the class of work for which particular health and working conditions are most likely.

## Figures and Tables

**Figure 1 ijerph-16-05080-f001:**
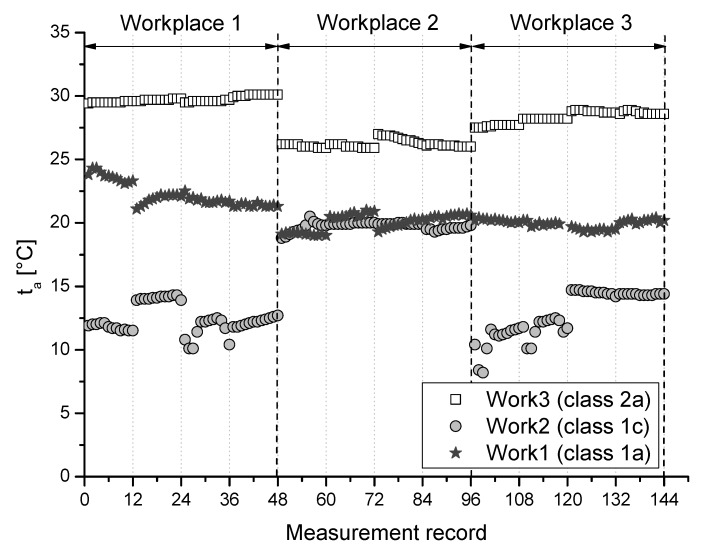
Air temperature at the monitored workplaces for individual class of work.

**Figure 2 ijerph-16-05080-f002:**
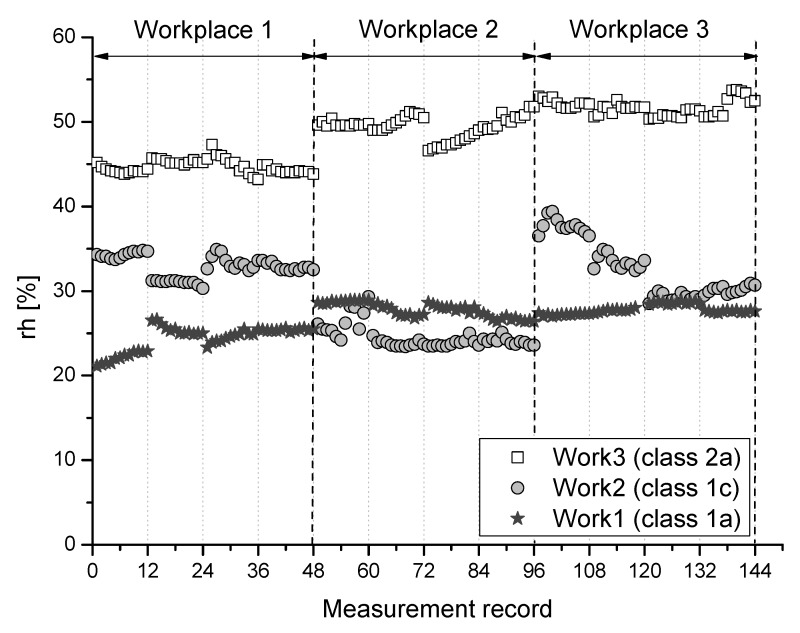
Relative humidity at the monitored workplaces for individual class of work.

**Figure 3 ijerph-16-05080-f003:**
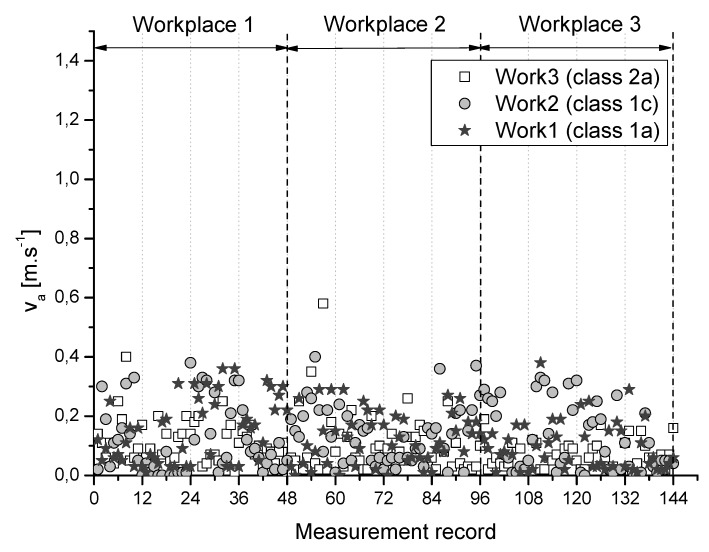
Airflow velocity at the monitored workplaces for individual class of work.

**Figure 4 ijerph-16-05080-f004:**
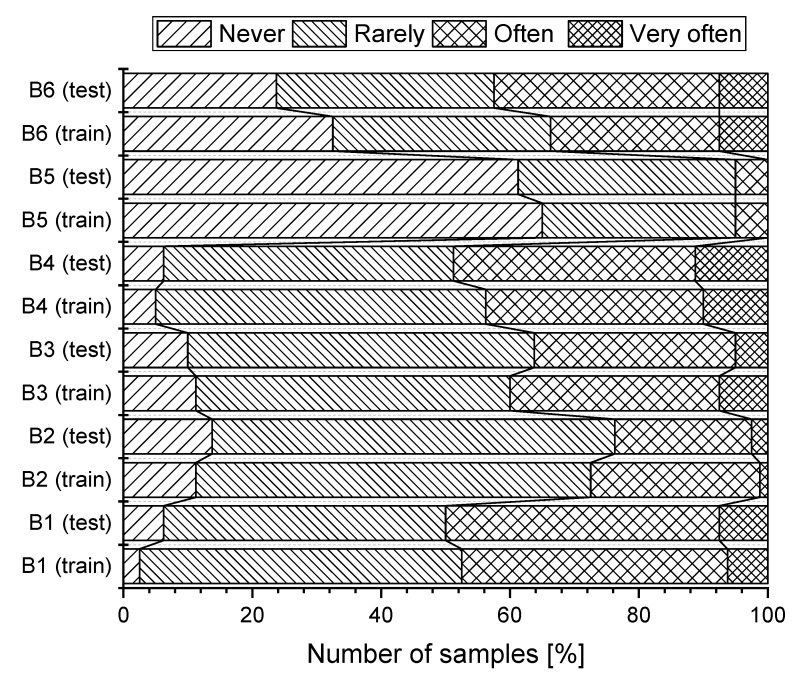
Heat and humidity conditions from the perspectives of the training and testing group.

**Figure 5 ijerph-16-05080-f005:**
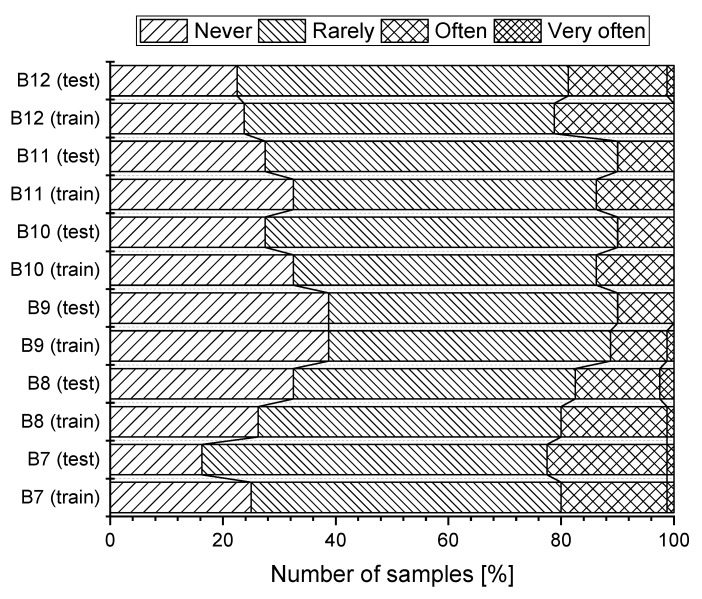
Health symptoms of employees in the training and testing group.

**Figure 6 ijerph-16-05080-f006:**
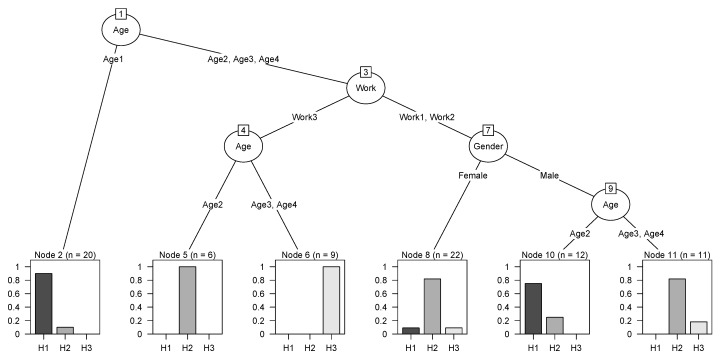
Decision tree (H1-Health1, H2-Health2, H3-Health3), (Output: software R).

**Table 1 ijerph-16-05080-t001:** Pearson Chi-squared test of independence (α=0.05, n=160).

Variables		B1	B2	B3	B4	B5	B6
**A1**	χ2-test	8.36	15.21	11.40	2.06	0.84	8.64
	*p*-value	**0.039 ***	**0.002 ***	**0.010 ***	0.357	0.656	0.034
	$C_{p}$	0.223	0.295	0.258	–	–	–
**A2**	χ2-test	15.15	11.71	18.43	16.22	7.21	18.87
	*p*-value	0.087	0.069	**0.005 ***	0.062	0.320	0.026
	$C_{p}$	–	–	0.321	–	–	–
**A3**	χ2-test	4.73	9.02	4.24	14.41	21.27	17.36
	*p*-value	0.316	0.061	0.375	**0.006 ***	**0.0001 ***	**0.008 ***
	$C_{p}$	–	–	–	0.287	0.343	0.313

Note: * *p*-value <α, A1—gender of the respondent, A2—age, A3—class of work, B1—satisfaction with the humidity in the room, B2—increased humidity in the room, B3—satisfaction with the airflow in the room, B4—satisfaction with the air temperature, B5—increase in the air temperature, B6—evaluation of feeling related to thermal conditions.

**Table 2 ijerph-16-05080-t002:** Pearson Chi-squared test of independence (α=0.05, n=160).

Variables		B7	B8	B9	B10	B11	B12
**A1**	χ2-test	3.64	3.75	6.06	0.60	5.79	1.59
	*p*-value	0.16	0.154	**0.048 ***	0.740	0.055	0.451
	$C_{p}$	–	–	0.191	–	–	–
**A2**	χ2-test	32.70	35.52	39.51	65.82	31.27	28.26
	*p*-value	**0.0001 ***	**0.0001***	**0.0001 ***	**0.0001 ***	**0.0001 ***	**0.0001 ***
	$C_{p}$	0.412	0.426	0.445	0.540	0.404	0.387
**A3**	χ2-test	0.84	2.13	7.13	3.05	8.42	2.22
	*p*-value	0.933	0.712	0.129	0.550	0.077	0.695
	$C_{p}$	–	–	–	–	–	–

Note: * *p*-value <α, A1—gender of the respondent, A2—age, A3—class of work, B7—feeling tired, B8—having pain in the spine, B9—headache, B10—feeling colds, B11—feeling of dried nasal mucosa and B12—the incidence of health problems.

**Table 3 ijerph-16-05080-t003:** Description of the investigated variables.

Variable	Description
*Input*	
Gender (A1)	2 classes (male, female)
Age (A2)	4 classes (Age1—up to 30 years, Age2—from 31-40 years, Age3—from 41–50 years, Age 4—over 50 years)
Work (A3)	3 classes (Work1—class 1a, Work2—class 1c, Work3—class 2a)
*Output*	
Health (y)	3 classes (Health1—excellent health, without serious health problems, Health2—average health, mild or moderate health problems, respectively, Health3—poor health, serious health problems)

**Table 4 ijerph-16-05080-t004:** Confusion matrix for three classes of the output variable Health (training group).

Observed Health Status	Model-Determined Classification		
	Health1	Health2	Health3
Health1	27	2	0
Health2	5	33	0
Health3	0	4	9

**Table 5 ijerph-16-05080-t005:** Confusion matrix for the three classes of the output variable Health (test group).

Observed Health Status	Model-Determined Classification		
	Health1	Health2	Health3
Health1	24	3	0
Health2	4	37	0
Health3	0	5	7

**Table 6 ijerph-16-05080-t006:** The level of correlation of classification.

Group	Overall Accuracy	Cohen’s kappa	95% Confidence Intervals for Cohen’s kappa	Strength of Agreement
Training	0.86	0.77	(0.648, 0.898)	Good
Test	0.85	0.74	(0.607, 0.876)	Good

**Table 7 ijerph-16-05080-t007:** The final classification model.

Age	Gender	Class of Work		
		Class 1a	Class 1c	Class 2a
Up to 30 years	Male	Health1	Health1	Health1
	Female	Health1	Health1	Health1
from 31 to 40 years	Male	Health1	Health2	Health2
	Female	Health2	Health2	Health2
from 41 to 50 years	Male	Health2	Health2	Health2
	Female	Health2	Health2	Health3
over 50 years	Male	Health2	Health2	Health3
	Female	Health2	Health2	Health3

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
