# Peer review of "Analysis of the Impact of Selected Physical Environmental Factors on the Health of Employees: Creating a Classification Model Using a Decision Tree"

_ijerph, 2019, doi:10.3390/ijerph16245080_

Round 1

Reviewer 1 Report

In Introduction, lines 57 and 61 are only present the reference without indicating the name of the main author in the text.

In material and methods: Line 113 describes other variables included, ending with "...and so on". It is suggested to describe all the variables or questions included and not leave room for the reader's imagination.

It would be advisable to describe the type of companies in which the study was conducted, productive sector, industries? services? Offices? outdoor work? Although it has no direct implications on the study and its results, it is of interest to better understand the context.

In results: Tables 1 and 2 show the variables with numbers (A1-A2-A3, B1 ... B6). It would be useful to include a table with the description of each one (it is in the text, but when you see the table, the description is on another page and one is lost). Another possibility is to put the variables directly in words (age, sex, etc.).

There is no description of the limitations of the study, as there is no discussion chapter and the discussion is incorporated with the results. The point of limitations is lost. I suggest including a subtitle for limitations.
For example, nothing says about the influence of the use of personal protection elements, training in its use. Also, the researchers assumed that employers respect sanitary conditions in the work environment. This is important when playing a potential role as confusers, especially in the influence on the state of health that is the outcome of the study.

Author Response

Response to Reviewer 1 Comments

First of all, we want to thank you very much for your valuable advice and comments. Based on your feedback and suggestions, we've made the following changes:

Your response 1: „In the Introduction, lines 57 and 61 are only present the reference without indicating the name of the main author in the text.“

Answer 1: We added the main authors in Lines from 59 to 60  and after a new one 64 to the cited references.

Your response 2: „In material and methods: Line 113 describes other variables included, ending with "...and so on". It is suggested to describe all the variables or questions included and not leave room for the reader's imagination.“

Answer 2: After the new one in Line 115, we added "time to rest" and ended the sentence.

Your response 3: „It would be advisable to describe the type of companies in which the study was conducted, productive sector, industries? services? Offices? outdoor work? Although it has no direct implications on the study and its results, it is of interest to better understand the context.“

Answer 3: We added the following sentences in Lines 92-95: „The companies, in which the measurements of microclimate parameters were performed and subsequently the questionnaire survey, were industrial plants engaged in the production of components for cars. All employees, that were subject to research, worked indoors.“

Your response 4: „In results: Tables 1 and 2 show the variables with numbers (A1-A2-A3, B1 ... B6). It would be useful to include a table with the description of each one (it is in the text, but when you see the table, the description is on another page and one is lost). Another possibility is to put the variables directly in words (age, sex, etc.).“

Answer 4: Below the Tables 1 and 2, we added an explanation of each variable to note.

Your response 5: „There is no description of the limitations of the study, as there is no discussion chapter and the discussion is incorporated with the results. The point of limitations is lost. I suggest including a subtitle for limitations. For example, nothing says about the influence of the use of personal protection elements, training in its use. Also, the researchers assumed that employers respect sanitary conditions in the work environment. This is important when playing a potential role as confusers, especially in the influence on the state of health that is the outcome of the study. „

Answer 5: We have added a discussion chapter that we have divided into other subchapters, namely "Use of decision trees", "Effects of inappropriate microclimate conditions" and "Prevention, measures, and limitations". We have added texts to this chapter and also moved some texts from the Introduction and Results, that is more appropriate to the Discussion chapter.

Reviewer 2 Report

The authors studied an innovative method using classification tree methodology for the occupational world. The methodology is described well, examples of previous work with these methods is described and the findings support the publication of this article. I have some comments that can be used to improve the paper.

I don't follow the reasoning for having lines 40-60. Please describe the reason for this in a topic sentence at the beginning of the paragraph. If the paragraph starting in line 31 sets up the next paragraph, please provide a better transition between the two. For example, a sentence such as line 80-81 would help.

Line 61: Missing citation callout for #13.

Line 98: Describe the Testo 435 instrument better (manufacturer, instrument parameters)

Lines 125-135: Is this more suitable for the Results section?

Discussion section is missing. This section should be used to link your findings to the findings of other studies. I understand this is the first study to use decision trees for occupational-related issues, but the authors should provide some context of how their findings compare to others works using decision trees.

I only saw one stated limitation of this study (line 345). Please state other limitations (including those pertaining to the methodology). 

Author Response

Response to Reviewer 2 Comments

First of all, we want to thank you very much for your valuable advice and comments. Based on your feedback and suggestions, we've made the following changes:

Your response 1: „I don't follow the reasoning for having lines 40-60. Please describe the reason for this in a topic sentence at the beginning of the paragraph. If the paragraph starting in line 31 sets up the next paragraph, please provide a better transition between the two. For example, a sentence such as line 80-81 would help.”

Answer 1: To the lines 40 and 41, we added the following sentence: „Evaluation of the working environment from the viewpoint of microclimatic conditions is an important topic, which is of great interest from the research point of view.“ This sentence describes the reason for this paragraph and creates a better transition between the paragraphs.

Your response 2: „Line 61: Missing citation callout for #13.”

Answer 2: We added the main author, after a new one in Line 64 and callout for #13, to the cited reference.

Your response 3: „Line 98: Describe the Testo 435 instrument better (manufacturer, instrument parameters)”

Answer 3: To the Lines, from 96 to 101, we added the manufacturer of the Testo 435-2 instrument and also some technical parameters, as follows: „These parameters were measured by the multifunctional Testo 435-2 instrument with the hot wire probe (Ø 12 mm) for measurements of air velocity, temperature, and relative humidity. The temperature measuring range -20 to +70°C (accuracy ±0.3°C), the relative humidity measuring range 0 to +100% (accuracy ±2%) and the air velocity measuring range 0 to +20m/s (accuracy ±0.03 m/s). The manufacturer is the company TESTO GmbH.“

Your response 4: „Lines 125-135: Is this more suitable for the Results section?”

Answer 4: Thank you for your feedback. We moved this paragraph to the chapter "Results and findings", Lines from 171 to 181.

Your response 5: „Discussion section is missing. This section should be used to link your findings to the findings of other studies. I understand this is the first study to use decision trees for occupational-related issues, but the authors should provide some context of how their findings compare to others works using decision trees.“

„I only saw one stated limitation of this study (line 345). Please state other limitations (including those pertaining to the methodology).“

Answer 5: We have added a discussion chapter that we have divided into other subchapters, namely "Use of decision trees", "Effects of inappropriate microclimate conditions" and "Prevention, measures, and limitations". We have added texts to this chapter and also moved some texts from the Introduction and Results, that is more appropriate to the Discussion chapter.

This manuscript is a resubmission of an earlier submission. The following is a list of the peer review reports and author responses from that submission.